# Mitochondrial DNA Damage and Its Repair Mechanisms in Aging Oocytes

**DOI:** 10.3390/ijms252313144

**Published:** 2024-12-06

**Authors:** Hiroshi Kobayashi, Shogo Imanaka

**Affiliations:** 1Department of Gynecology and Reproductive Medicine, Ms.Clinic MayOne, 871-1 Shijo-cho, Kashihara 634-0813, Japan; shogo_0723@naramed-u.ac.jp; 2Department of Obstetrics and Gynecology, Nara Medical University, 840 Shijo-cho, Kashihara 634-8522, Japan

**Keywords:** aging oocytes, mitochondrial DNA (mtDNA), mtDNA mutations, nicotinamide adenine dinucleotide, poly ADP-ribose polymerase 1 (PARP1)

## Abstract

The efficacy of assisted reproductive technologies (ARTs) in older women remains constrained, largely due to an incomplete understanding of the underlying pathophysiology. This review aims to consolidate the current knowledge on age-associated mitochondrial alterations and their implications for ovarian aging, with an emphasis on the causes of mitochondrial DNA (mtDNA) mutations, their repair mechanisms, and future therapeutic directions. Relevant articles published up to 30 September 2024 were identified through a systematic search of electronic databases. The free radical theory proposes that reactive oxygen species (ROS) inflict damage on mtDNA and impair mitochondrial function essential for ATP generation in oocytes. Oocytes face prolonged pressure to repair mtDNA mutations, persisting for up to five decades. MtDNA exhibits limited capacity for double-strand break repair, heavily depending on poly ADP-ribose polymerase 1 (PARP1)-mediated repair of single-strand breaks. This process depletes nicotinamide adenine dinucleotide (NAD⁺) and ATP, creating a detrimental cycle where continued mtDNA repair further compromises oocyte functionality. Interventions that interrupt this destructive cycle may offer preventive benefits. In conclusion, the cumulative burden of mtDNA mutations and repair demands can lead to ATP depletion and elevate the risk of aneuploidy, ultimately contributing to ART failure in older women.

## 1. Introduction

Our female hunter–gatherer ancestors devoted a significant portion of their reproductive years to pregnancy and lactation [1]. However, they were also subject to the adverse health outcomes associated with excessive pregnancies, including maternal and fetal mortality during childbirth. In developed nations, advancements in contemporary perinatal care have markedly diminished life-threatening complications linked to pregnancy and childbirth; however, these risks persist at alarming levels in developing regions. In recent years, societal advancements have driven a global trend toward delayed marriage and later onset of first pregnancies [2]. The continuous physiological demand of monthly ovulation can negatively impact women’s health, contributing to reduced fecundity due to reproductive aging and an increased risk of conditions such as uterine fibroids, endometriosis, and gynecological cancers [3]. Notably, the age-related decline in ovarian reserve significantly impairs reproductive potential [4,5]. Assisted reproductive technologies (ARTs) for women with aging ovaries are less effective and entail greater financial burdens [6]. For instance, the live birth rate per oocyte stands at 26% for women under 35 but drops to merely 1% for those over 42 [7].

Although the mechanisms underlying reproductive aging remain incompletely understood, it is conceptually attributed to mitochondrial dysfunction, which may induce oxidative stress, hormonal dysregulation, reduced oocyte quantity and quality, mitochondrial DNA (mtDNA) damage and related genetic mutations, epigenetic alterations, defective spindle assembly, meiotic errors, chromosomal misalignment, and telomere shortening [8,9,10,11,12]. Oocyte quantity and quality may decline with age, potentially contributing to impaired mitochondrial function [12], or the relationship may be bidirectional, with both influencing each other [11,13,14]. Furthermore, granulosa and cumulus cells are known to promote oocyte maturation by activating glycolysis [15], the pentose phosphate pathway [16], and lipid metabolism [13,17]. Notably, this bidirectional regulatory mechanism is crucial, as mature oocytes, in turn, stimulate and activate glycolysis and the tricarboxylic acid (TCA) cycle in granulosa cells [13,18]. A recent metabolomic analysis of the aging ovary revealed diminished mitochondrial oxidative phosphorylation (OXPHOS) despite compensatory upregulation of glycolysis and lipid metabolism [19]. Impaired OXPHOS results in reduced adenosine 5′ triphosphate (ATP) production, a critical factor since mitochondria serve as the primary energy source during oocyte maturation, fertilization, and early embryonic development [20]. The segregation of chromosomes to opposite poles during meiosis demands substantial energy [14,21]. Age-associated mutations in mtDNA are believed to induce ATP depletion, resulting in errors in chromosome segregation [11,22].

In this review, we first provide a brief summary of our current understanding of age-related mitochondrial changes and ovarian aging. We subsequently present an updated overview of the causes and mechanisms underlying mitochondrial DNA mutations and their potential repair, concluding with a discussion of prospective therapeutic strategies to alleviate the impact of aging on oocytes.

## 2. Normal Mitochondrial Function

Oxygen levels in Earth’s atmosphere rose dramatically between 2.4 and 2 billion years ago, a period known as the “Great Oxidation Event” [23]. This transformative event was driven by cyanobacteria, which released substantial amounts of oxygen through photosynthesis, creating conditions that were pivotal for the evolution of life and enabling the emergence of complex organisms [23]. Approximately two billion years ago, bacteria capable of using oxygen for energy production were incorporated into the ancestors of eukaryotic cells, evolving into mitochondria—organelles essential for energy metabolism in modern eukaryotic cells [24]. Mitochondria, often referred to as the “powerhouses of the cell”, are central to bioenergetics and metabolic processes, such as oxidative phosphorylation, and exhibit dynamic transformations to meet fluctuating cellular energy demands [6,25,26]. ATP is synthesized via the electron transport chain, which consists of five complexes embedded in the inner mitochondrial membrane [6,27]. Complexes I (nicotinamide adenine dinucleotide (NADH) dehydrogenase) and II (succinate dehydrogenase) oxidize NADH and flavin adenine dinucleotide (FADH2), respectively, transferring electrons to ubiquinone (coenzyme Q10, CoQ10) [28] (Figure 1). The NADH and FADH2 utilized by these complexes are generated through the mitochondrial TCA cycle, underscoring the intricate coordination between the TCA cycle and the electron transport chain in facilitating ATP production [27]. Complex III reduces cytochrome c, which subsequently donates electrons to molecular oxygen, enabling electron transfer to Complex IV [6,27,28]. Complex V (ATP synthase) drives ATP synthesis by utilizing the proton gradient generated across the inner membrane [28]. However, interactions between the electron transport chain and molecular oxygen generate superoxide anion radicals (O_2_^•−^), which dismutate into hydrogen peroxide (H_2_O_2_) and can further react to produce hydroxyl radicals (HO^•^) [25,29,30,31,32]. Reactive oxygen species (ROS), as by-products of mitochondrial respiration, function as critical signaling molecules in preserving the health and homeostasis of organisms [33]; however, excessive ROS production generates cytotoxic and mutagenic free radicals, posing significant risks [6,26,28]. Thus, the “endosymbiotic contract” reflects a delicate equilibrium between the evolutionary advantages mitochondria confer and the intrinsic challenges posed by the reactive by-products of aerobic respiration, shaping the complex dynamics of multicellular life [24]. Young mitochondria, depicted in the center of Figure 1, exhibit enhanced ATP synthesis and attenuated ROS production. Conversely, aged mitochondria, shown on the right, display impaired ATP synthesis, mitochondrial dysfunction, elevated ROS production, and consequent mitochondrial DNA damage. Thus, mitochondria in younger cells predominantly harbor wild-type mtDNA, whereas those in aged cells exhibit a higher proportion of mutated mtDNA.

## 3. Age-Related Changes in Mitochondria

As women age, both the quantity and quality of oocytes diminish, undermining their developmental competence post-fertilization and elevating the risk of miscarriage [34,35]. However, when women over 40 undergo in vitro fertilization (IVF) with oocytes donated by younger women, success rates align with those of younger patients, indicating that the donor oocyte quality—rather than the recipient’s age—is pivotal in determining IVF outcomes [34,36]. Age-related morphological and ultrastructural alterations in oocytes have been documented [37,38,39]. Pronounced mitochondrial ultrastructural differences between younger and older women include variations in mitochondrial size and shape, cristae architecture, membrane integrity, and mitochondrial density [11,39,40]. In older women, both mitochondrial quantity and density are reduced, with the inner membranes forming the cristae appearing disorganized [38]. Mitochondria frequently exhibit enlargement, elongation, and irregular shapes, while younger women typically display mitochondria with a more consistent and healthy morphology [38]. These ultrastructural changes in older women are often linked to a decline in mitochondrial membrane integrity and compromised mitochondrial function [37].

Furthermore, the impact of aging on oocyte mitochondria is evident through functional abnormalities [6,11]. Age-associated biochemical changes encompass alterations in glucose and energy metabolism, mitochondrial function, lipid metabolism, oxidative stress, amino acid and protein metabolism, calcium signaling, steroid hormone signaling, gap junction communication, paracrine signaling, and epigenetic modifications [6,11]. Mitochondrial dysfunction, in particular, involves several key aspects, including diminished mitochondrial content, reduced mitochondrial protein synthesis, impaired antioxidant activity, defects in mitochondrial dynamics (such as biogenesis, fusion and fission, mitophagy (selective degradation of damaged mitochondria), or apoptosis) [41], metabolic imbalances, disrupted mitochondrial membrane potential (Δψ) [42], dysfunctional electron transport chain activity, decreased ATP production [43], elevated ROS levels, and activation of the mitochondrial permeability transition pore [6,11,25,44,45]. The most critical consequence of age-related mitochondrial dysfunction is impaired ATP synthesis and mtDNA damage, which negatively influences chromosome segregation [46,47] and compromises embryonic development [48,49,50] (Figure 1). Impaired chromosome segregation during meiosis leads to elevated rates of aneuploidy [22], and there is little doubt that mitochondrial dysfunction is a key contributor to ovarian aging.

## 4. Ovarian Ageing as the Free Radical Theory

A substantial body of research has focused on the complex interplay between aging, mitochondrial function, and oxidative stress in oocytes [12,51,52]. The widely accepted free radical theory [25,29] serves as the prevailing hypothesis to explain aging, positing that ROS drive the process of cellular senescence [53,54,55,56]. Some researchers emphasize that repeated ovulation and menstrual cycles, along with other sources of ROS accumulation in the ovaries—such as obesity and unhealthy lifestyle factors—impose stress on the reproductive system, resulting in excessive oxidative stress [30]. Numerous studies revealed that elevated oxidative stress has been shown to impair mitochondrial function, thereby compromising oocyte quality and fertility [45,57,58,59,60].

Let us now examine the mechanism through which oxidative stress induces damage to mtDNA. Arbeithuber et al. conducted an analysis of mtDNA mutations in 30 Indian rhesus macaques, ranging in age from 1 to 23 years, stratified into four distinct age groups: <5 years, 5–10 years, 10–15 years, and 15–23 years [61]. Their findings revealed that de novo mtDNA mutations progressively accumulate with advancing age, predominantly manifesting as transitions rather than transversions [61]. Transition mutations, characterized by purine-to-purine or pyrimidine-to-pyrimidine substitutions, are less disruptive to DNA structure [62]. In contrast, transversion mutations, which involve purine-to-pyrimidine exchanges (or vice versa), are more likely to impair DNA structure and function [62,63]. Evidence suggests that the majority of mtDNA mutations associated with aging are transitions, whereas ROS-induced mutations are predominantly transversions [64]. The dominance of transitions over transversions in mtDNA mutations is noteworthy, as it highlights the biochemical processes underlying mutagenesis. Transitions are primarily linked to replication errors or repair mechanisms that favor nucleotide mispairing within the same chemical class [64]. Therefore, it is hypothesized that mtDNA replication errors become increasingly frequent with advancing age. On the other hand, ROS primarily induce transversion mutations, such as the conversion of guanine to 8-oxoguanine, which pairs with adenine and results in a G → T transversion [63]. This indicates that mitochondria possess an efficient capacity to repair mtDNA damage induced by oxidative stress [65]. Outlined below are the molecular mechanisms through which oxidative stress contributes to oocyte aging: Oxidative stress induces DNA damage and mtDNA mutations (resulting in chromosomal instability and elevated aneuploidy) [66], protein oxidation (leading to impaired protein function and aggregation), lipid peroxidation (compromising cell membranes and amplifying oxidative injury) [67], cytoskeletal damage (disrupting meiotic spindle integrity and causing chromosomal segregation errors), telomere shortening (accelerating cellular aging and oocyte senescence) [68], dysregulation of redox homeostasis (reducing antioxidant defenses and increasing oxidative damage), and epigenetic alterations (modifying gene expression and impairing developmental potential) [69,70,71]. Many researchers posit that the free radical hypothesis constitutes one of the primary triggers for mtDNA mutations. A detailed discussion of this hypothesis is beyond the scope of this review. Refer to sources [25,29] for further information.

## 5. Molecular Mechanism Underlying Oocyte Aging Caused by mtDNA Mutations

Here, we summarize the role of mitochondria in oocyte aging, with an emphasis on mtDNA mutations. Normal respiratory chain function requires a sufficient number of intact and functional mitochondrial genomes [72]. Each mitochondrion harbors approximately 2–10 copies of mtDNA [73]. Human mtDNA is a circular, double-stranded genome consisting of 16,569 bp, encoding only 13 proteins essential for the respiratory chain, while the remaining ~1500 mitochondrial proteins are encoded by the nuclear genome [26,32,72,74,75]. In primordial germ cells, mtDNA copy numbers are relatively low (~200 copies), but during oocyte maturation, mtDNA copy numbers expand dramatically, exceeding 200,000 in fully mature oocytes [36,76,77,78,79,80]. This amplification ensures the availability of sufficient mitochondria and energy reserves to support early embryonic development post-fertilization [76]. Conversely, sperm cells contain only 10 to 1000 mtDNA copies [81,82]. After fertilization, sperm mitochondria are degraded, and the embryo inherits its mitochondrial population exclusively from the maternal lineage [6,36,83]. MtDNA mutations in oocytes impede their ability to effectively support early embryonic development. The key mechanisms by which mtDNA mutations occur are ROS damage, replication errors, mitochondrial DNA repair deficiencies, replicative segregation and genetic drift, age-related accumulation of mutations, and environmental factors (Figure 2). Section 5.1, Section 5.2, Section 5.3, Section 5.4, Section 5.5 and Section 5.6 in the text below correspond to numbers 1 through 6 in Figure 2. Given the limited understanding of mitochondrial DNA damage and its role in oocyte aging, we will provide only a concise discussion on this topic from 5.3 onwards.

### 5.1. ROS-Induced Damage

ROS can directly damage mtDNA, causing base modifications, strand breaks, and DNA cross-linking. Specific base changes, such as guanine’s conversion to 8-oxoguanine, can result in GC-to-TA transversions, contributing to chromosomal instability [84]. ROS-induced damage can affect complex I (NADH dehydrogenase), impair NAD+ production, and reduce ATP synthesis [11,85]. NAD+/NADH redox pairs are central not only to OXPHOS but also to glycolysis, the tricarboxylic acid cycle, and fatty acid oxidation [85,86]. mtDNA mutations impair the redox balance of NADH and NAD+ [11,85]. For more information on age-related mtDNA mutations, see Section 7.

### 5.2. Replication Errors

Mitochondria replicate their DNA independently of the cell cycle, increasing the likelihood of replication errors [72]. POLG, responsible for mtDNA replication, is less efficient and accurate than nuclear DNA polymerases [87]. Despite having proofreading abilities, POLG can still introduce point mutations, insertions, or deletions. Replication can stall due to secondary structures or damage, causing strand breaks and rearrangements. Moreover, POLB plays a direct role in repairing oxidative lesions, such as 8-oxoguanine (8-oxoG), through its polymerase activity, ensuring mtDNA integrity [65]. POLB contributes high-fidelity nucleotide incorporation during repair. This prevents mutagenic events that could arise from errors in repair synthesis. In addition to single-nucleotide repair, POLB can participate in long-patch BER, where a stretch of 2–10 nucleotides is replaced. Refer to Section 8 for a comprehensive overview of the mechanisms involved in mtDNA damage repair.

### 5.3. Mitochondrial DNA Repair Deficiencies

Mitochondria possess limited DNA repair mechanisms, primarily base excision repair and mismatch repair [88]. However, mitochondrial MMR is relatively weak and may not provide substantial protection against the accumulation of mutations. Furthermore, they lack advanced repair pathways, such as nucleotide excision repair and homologous recombination, making mtDNA particularly susceptible to accumulating mutations [88]. ROS-induced double-strand breaks are especially detrimental, often leading to large deletions or rearrangements.

### 5.4. Replicative Segregation and Genetic Drift

When the mitochondrial population within an individual predominantly contains wild-type mtDNA, the condition is referred to as homoplasmy [6,89]. However, individual cells typically harbor a mixture of mutated and wild-type mtDNA, known as heteroplasmy [6,89,90]. During cell division, mtDNA is randomly partitioned among daughter cells in a process called replicative segregation [91]. With successive divisions, this stochastic distribution can lead to fluctuations in the ratio of mutated to non-mutated mtDNA across different cells. This stochastic distribution can result in the fixation (heteroplasmy) or elimination (homoplasmy) of specific mutations through genetic drift, particularly in cells with fewer mitochondria. Genetic drift describes the random variation in allele frequencies within a population, driven by chance rather than natural selection. Over time, random segregation increases the mutational load in some cells, exacerbating mitochondrial dysfunction. Refer to Section 6 for more information on heteroplasmy.

### 5.5. Age-Related Accumulation of Mutations

As mitochondria continuously divide and replicate throughout an individual’s lifetime, replication errors accumulate, contributing to the gradual buildup of mtDNA mutations [92]. DNA replication errors can lead to the emergence of transition mutations. The efficiency of mitochondrial DNA repair mechanisms also declines with age, accelerating the accumulation of damage. Consequently, mtDNA accumulates mutations at a higher rate than nuclear DNA over time.

### 5.6. Environmental Factors

Several environmental factors can induce mtDNA mutations by causing damage to the mitochondrial genome or disrupting its replication and repair processes. These factors include ultraviolet radiation, chemical toxins (pollutants, pesticides, tobacco smoke, and iron), radiation exposure, drugs and pharmaceuticals (chemotherapeutic agents, antibiotics, and endocrine-disrupting chemicals), and dietary and lifestyle factors (poor diet and alcohol consumption) [93]. Each of these factors can exacerbate the natural accumulation of mtDNA mutations over time, potentially contributing to aging and various mitochondrial and systemic diseases.

Collectively, increased mtDNA mutations, deletions, and respiratory chain deficiencies are key factors in oocyte aging [6,69,94,95,96,97]. MtDNA mutations arise from a combination of oxidative stress, replication errors, and limited repair capacity. ROS-induced damage may be a primary driver, given the oxidative environment within mitochondria. The accumulation of mutations over time, along with errors introduced by POLG during replication and the randomness of replicative segregation, renders mtDNA highly prone to mutations. Aneuploidy, a hallmark of aging, arises from DNA damage, loss of chromosomal cohesion, spindle assembly checkpoint dysfunction, and meiotic recombination errors [12,22,98]. These mutations are especially detrimental in energy-demanding cells like oocytes, contributing to age-related functional decline.

## 6. Heteroplasmy as a mtDNA Variant

In the fetal ovary, oocytes originate from primordial germ cells and remain arrested at the diplotene (germinal vesicle) stage of the first meiotic division until puberty [21,99,100]. Upon stimulation by a luteinizing hormone surge, fully grown oocytes resume development, complete the first meiotic division, and extrude the first polar body [21,100]. Subsequently, oocytes arrest at the metaphase of the second meiotic division until fertilization occurs [21,100]. During the prolonged arrest, lasting up to 50 years [6], mtDNA is susceptible to accumulating damage or replication errors, which may result in heteroplasmy or mtDNA deletions [61]. Despite the presence of multiple mtDNA mutations in oocytes, these are not frequently transmitted to offspring. Heteroplasmy is ubiquitous, with the average individual carrying at least one heteroplasmic variant [83,101,102]. Oocytes have developed mechanisms to minimize the intergenerational transmission of deleterious mtDNA mutations, a concept encapsulated by the bottleneck theory. The genetic bottleneck in mtDNA refers to the sharp reduction in mtDNA copy number during embryonic oogenesis, followed by amplification in subsequent stages [27,76,80,82,83,103,104,105]. While one consequence of this process is a reduction in the transmission of deleterious mtDNA mutations, the primary purpose and theoretical basis of the bottleneck extend beyond mutation reduction. The bottleneck amplifies stochastic differences in the proportions of mutant and wild-type mtDNA. This amplifies selection opportunities. If deleterious mutations dominate the reduced pool, these oocytes are less likely to develop or be viable. This is because irreparable mtDNA mutations are removed through mitophagy [106]. If wild-type mtDNA predominates, these oocytes have a better chance of propagating. A newly acquired mtDNA mutation is not immediately detrimental, as the wild-type genomes can compensate for the defect [107]. Consequently, low levels of mutant mtDNA generally do not impair offspring viability [108]. These mechanisms play a pivotal role in eliminating defective mitochondrial genomes and maintaining mitochondrial integrity across generations [6,108]. Furthermore, by introducing a degree of randomness in the segregation of mtDNA variants, the bottleneck increases genetic variability among offspring. This variability may have evolutionary advantages, allowing populations to adapt better to environmental changes or pressures. However, advanced maternal age at the time of fertilization correlates with an increased burden of heterozygous mtDNA mutations in the offspring’s blood and buccal cells [90]. The observed correlation between advanced maternal age at fertilization and an elevated number of heteroplasmies in the offspring implies that the transmission frequency of mutations increases with maternal age [90,109]. Therefore, the timing of mtDNA mutations is associated with the duration preceding the meiotic division; the longer this period, the higher the likelihood of mtDNA mutations accumulating. As shown in Figure 3 on the left, the extent and distribution of mtDNA mutations vary across individual mitochondria. It is postulated that there are various types, ranging from type A, which exhibits few mutations, to type B, which has moderate mutations, to type C, characterized by numerous mutations. It has been reported that mitochondrial dysfunction typically becomes apparent only when the mutation rate exceeds 80% within a heteroplasmic cell [110]. For example, in types A, B, and C, the mtDNA mutation rates are 33%, 67%, and 100%, respectively. While type C exhibits mitochondrial dysfunction, types A and B do not, despite the presence of mtDNA mutations (Figure 3, left). Thus, the proportion of type C mitochondria within an oocyte may dictate the destiny of the ovarian reserve in older women.

## 7. Age-Related Alterations in mtDNA Mutations and Copy Number

Here, we provide a summary of the age-associated dynamics in mtDNA mutations and copy number. Mutations were detected in the mtDNA of 28% of oocytes and 66% of cumulus cells, demonstrating a markedly lower mutation burden in germ cells relative to somatic cells [111,112]. On the other hand, the incidence of mtDNA mutations in sperm is exceptionally high [113,114]. Common mtDNA mutations include point mutations [115] and the widely recognized 4977 bp deletion—often termed the “common deletion” [116]. Research indicates that older women exhibit a significantly greater frequency of low-frequency point mutations in their oocytes than younger women [115]. Although fertilization rates appear comparable between older and younger women, blastocyst formation rates decline with maternal age [85]. This finding suggests that despite successful fertilization, oocytes from older women are less likely to develop into blastocysts [85]. In fact, mitochondrial function at the morula stage of human embryos deteriorates with maternal age, impairing the transition from morula to blastocyst [117]. The increased frequency of mtDNA mutations in older oocytes may deplete the energy reserves necessary for successful blastocyst progression. Furthermore, the 4977 bp deletion has significant repercussions for mitochondrial architecture and functionality [116]. The concomitant loss of numerous genes can disrupt ATP synthesis and compromise energy metabolism. Studies on cloned bovine oocytes suggest that mtDNA deletions increase with age [97], though human studies present conflicting evidence regarding the age-related accumulation of such deletions [116,118,119,120].

Next, it is well established that mtDNA content differs between the oocytes and embryos of younger and older women. Multiple studies indicate that oocytes from older women contain fewer mtDNA copies and a lower mtDNA content compared to those from younger women [6,73,116,121,122,123,124]. Similar trends have been reported in bovine models [6,125]. In contrast, embryos derived from older women exhibit an elevated mtDNA copy number relative to their younger counterparts [126]. This suggests that mitochondrial DNA in older women is more prone to mutations, necessitating an increased mtDNA copy number to meet the augmented ATP production demand [126]. This compensatory mechanism may account for the observed rise in mtDNA content in embryos from older women. Therefore, elevated mtDNA copy numbers have been correlated with reduced embryo quality and viability [127]. Indeed, embryos with increased mtDNA content may show diminished implantation potential during IVF [126,128,129,130,131]. Nonetheless, other studies report no significant association between mtDNA copy number and implantation outcomes [132,133] or between ovarian aging and the accumulation of mtDNA point mutations [111]. Although evidence remains conflicting, embryos from older women may display elevated mtDNA copy numbers; however, this increase alone is insufficient to guarantee a successful live birth without adequate energetic reserves to sustain blastocyst development.

## 8. mtDNA Damage and Its Repair Mechanism

We initially present a summary of the mechanisms governing mtDNA repair in oocytes (Figure 4). Among the mtDNA damage repair mechanisms, the BER pathway is critical for maintaining mtDNA integrity, addressing damage caused by ROS and other genotoxic insults [134,135]. The roles of Poly(ADP-ribose) polymerase (PARP) and polymerase in this process are as follows: PARP1/OGG1 binary complexes detect single-strand breaks (SSBs) or nicks caused by damaged bases that have been excised. Once activated, PARP1 catalyzes the addition of poly(ADP-ribose) (PAR) chains to itself and nearby proteins using NAD⁺ as a substrate [136,137]. PARP1 catalyzes the cleavage of NAD⁺ into nicotinamide and ADP-ribose. This modification recruits and stabilizes BER proteins at the site of damage, enhancing repair efficiency. Mitochondrial BER relies on specific DNA polymerases that perform vital roles in the repair mechanism. POLG is the primary mitochondrial DNA polymerase and plays a dual role in replication and repair. During BER, after the damaged base is excised by a glycosylase and the resulting abasic site is processed by an AP endonuclease, POLG fills the gap by incorporating the correct nucleotide. It also exhibits proofreading activity, ensuring high fidelity during gap filling. Emerging evidence suggests that additional polymerases, such as POLB, may transiently localize to mitochondria under stress [65]. Therefore, PARP activity and polymerase function are tightly regulated to maintain mitochondrial integrity. Dysfunction in these proteins, depletion of NAD⁺ levels, or introduction of replication errors can lead to impaired BER, mtDNA mutations, and mitochondrial dysfunction.

NAD⁺ synthesis involves ATP in several steps (the de novo pathway, the Preiss–Handler pathway, and the salvage pathway) [132]. ATP is essential for NAD⁺ synthesis and recycling, highlighting the interconnectedness of cellular energy metabolism and cofactor availability. If ATP production fails to meet the demand for NAD⁺ consumption over time, PARylation becomes compromised, preventing the completion of the mtDNA repair process. Therefore, chronic activation of mitochondrial PARP1 can deplete NAD⁺ and ATP, triggering mitochondrial dysfunction and eventual cell death (blue square in Figure 4) [138]. PARP1 enables effective mtDNA repair under physiological NAD⁺ concentrations but impairs polymerase activity when NAD⁺ is scarce [139]. These findings indicate that aging diminishes the capacity to repair mtDNA mutations.

Second, NAD⁺ also supports mitochondrial sirtuins, enzymes essential for metabolic regulation, genomic stability, DNA repair, inflammation control, and longevity [140]. Sirtuins modulate mitochondrial enzymes involved in DNA repair processes such as BER. Sirtuins catalyze lysine deacetylation, coupling it to NAD⁺ hydrolysis, generating O-acetyl-ADP-ribose and nicotinamide [141]. For example, SIRT1 has been proposed to enhance the expression of nuclear respiratory factor 1 (NRF-1), NRF-2, and mitochondrial transcription factor A (TFAM), a key protein responsible for stabilizing mtDNA and promoting its replication and transcription [142], via its deacetylation of peroxisome proliferator-activated receptor gamma coactivator 1-alpha (PGC-1α). Moreover, SIRT3 activates mitochondrial antioxidant enzymes like superoxide dismutase 2 (SOD2) and glutathione peroxidase, reducing oxidative damage to mtDNA [143]. This helps maintain the integrity of the mitochondrial genome. Additionally, sirtuins rely on NAD⁺ for their enzymatic activity, linking their function to cellular energy status and mitochondrial metabolism. Therefore, elevated NAD⁺ levels enhance sirtuin activity, promoting efficient mtDNA repair, replication, and transcription. Furthermore, sirtuins play a role in mitophagy [144], thereby removing mitochondria with extensively mutated or damaged mtDNA. Changes in NAD⁺ levels due to aging or stress directly affect sirtuin function, influencing health-span and cellular homeostasis. Beyond PARPs and sirtuins, NAD⁺ serves as a cofactor for other enzymes, including ADP-ribosyltransferase (ART), RNA polymerase, cyclic ADP-ribose hydrolase (CD38), and Sterile Alpha and TIR Motif Containing 1 (SARM1) [145]. These enzymes regulate metabolism and maintain intracellular equilibrium [145,146]. Excessive NAD⁺ depletion, resulting from metabolic stress, culminates in ATP deficiency and cellular death. The roles of NAD⁺ and sirtuins in oocyte protection and aging are gradually being elucidated.

Finally, we also outline the challenges in developing therapeutic strategies aimed at enhancing mtDNA repair. Since NAD+ is continuously expended to sustain intracellular homeostasis, oocytes necessitate perpetual replenishment to preserve diverse cellular functions [147,148]. Age-related diseases have been causally linked to declining NAD⁺ levels, and several preclinical interventions have demonstrated the benefits of NAD⁺ restoration [139,145,149,150]. Indeed, in aged mice, oocyte quality improved following NAD⁺ replenishment [139]. Administering nicotinamide riboside (NR), a NAD⁺ precursor, to mutant mice lacking critical NAD⁺ biosynthetic enzymes (indoleamine-2,3-dioxygenase 1 (Ido1) or quinolinate phosphoribosyl transferase (Qprt)) restored ovarian reserve and enhanced oocyte quality [151]. Similarly, nicotinamide mononucleotide (NMN) supplementation recovered NAD⁺ levels, improving oocyte quality and fertility in naturally aged mice [152]. A four-week regimen of NMN supplementation conferred significant benefits [139]. In a separate study, 40-week-old mice received NMN treatment for 20 weeks to validate its efficacy [153]. In mice, 1 month corresponds to approximately 2.5 human years, representing a relatively prolonged treatment duration for humans. ATP demand peaks during spindle assembly in metaphase I [154], necessitating adequate NAD⁺ replenishment for mtDNA repair [155]. The recent reviews from 2022 [147] and 2023 [145] offer comprehensive insights into NAD⁺ metabolism, ovarian aging, and the therapeutic promise of NAD⁺-boosting strategies. They have explored the roles of NAD⁺, sirtuins, and PARPs in fertility, with potential applications in embryo production programs [147]. However, the efficacy of NMN supplementation varied among species, with porcine models showing less pronounced benefits than mice [147]. The therapeutic efficacy of NAD⁺ precursor supplementation may vary based on local NAD⁺ concentrations.

The variability of mtDNA mutations and the efficacy of NAD⁺ precursor supplementation must be carefully considered. As mentioned in the previous subsection (Figure 3, left), the extent of mtDNA mutations can vary significantly among individual oocytes, even within a single organism. In aged mice harboring type A and type B mitochondria, NMN supplementation may partially ameliorate mtDNA mutations, promoting the predominance of wild-type mtDNA (Figure 3, right). This restoration of wild-type mtDNA may improve spindle assembly, cleavage rates, blastocyst formation, and live birth outcomes. However, in cases where type C mitochondria harbor numerous unrepairable mutations, NMN supplementation may not yield comparable benefits. Type B mitochondria, with intermediate levels of damage, may experience partial improvements following NMN administration. Therefore, NMN supplementation may not exert uniform effects across all animal models, as oocytes exhibit varying degrees of mitochondrial dysfunction.

## 9. Conclusions

In developed nations, there is a growing trend toward delayed pregnancy and childbirth, accompanied by an escalating demand for ART [139]. Advanced maternal age remains one of the most significant clinical challenges in reproductive medicine, yet effective strategies to enhance oocyte quality are not fully elucidated [139,152]. This review focuses primarily on mtDNA mutations and their repair mechanisms, following a concise discussion of the free radical theory. Age-related mtDNA mutations are believed to accumulate predominantly due to ROS-induced damage and deficiencies in repair pathways, ultimately leading to mitochondrial dysfunction [57,58]. The generation of ROS and mtDNA damage are intricately connected, with overlapping effects on mitochondrial health [145]. Quiescent oocytes—dormant for decades—are vulnerable to oxidative damage [69,145]. mtDNA is subject to damage, replication, and replication errors, resulting in heterogeneity and deletions [83,156]. Mitochondria harboring a high burden of mtDNA mutations respond by upregulating their mtDNA copy number through compensatory mechanisms. However, despite retaining fertilization potential, they remain inefficient at ATP production and fail to support blastocyst formation [85,117]. As mtDNA damage accumulates and repair capacity becomes overwhelmed or impaired, mutations continue to increase with age.

Studies have identified evolutionarily conserved mechanisms of mtDNA inheritance, such as bottlenecks during germ cell development and selective pressures against specific mtDNA mutations during maternal transmission [103,104,105]. Nonetheless, investigations into how aging affects mtDNA repair mechanisms are still in their early stages. Notably, mtDNA exhibits limited capacity to repair double-strand breaks, relying heavily on PARP1 for the repair of single-strand breaks [136,137]. However, this repair process consumes NAD⁺ as a substrate, potentially leading to ATP depletion over time, resulting in a paradox where oocyte function declines despite ongoing mtDNA repair. To preserve oocyte function with aging, it is crucial to establish an environment that ensures the continuous replenishment of NAD⁺ and ATP [145,147]. Nonetheless, the long-term administration of drugs—equivalent to 2.5 to 12.5 years in humans, based on animal studies—is not a practical solution [139,153]. In oocytes that have remained quiescent for decades, short-term antioxidant and NAD⁺ supplementation may be inadequate to correct pre-existing mtDNA mutations. Moreover, the mtDNA within individual mitochondria is highly heterogeneous, complicating repair efforts. While NAD⁺ supplementation may benefit mitochondria with relatively low mutation loads, those with a heavy mutation burden remain challenging to restore, perpetuating mitochondrial dysfunction. Therefore, the administration of NMN requires careful consideration of dosage, duration, delivery method, and the extent of pre-existing mtDNA mutations.

In conclusion, the accumulation of mtDNA mutations, coupled with the decline in repair mechanisms, contributes to reduced ovarian reserve and developmental competence, heightening the risk of aneuploidy. There is an urgent need for clinically viable and practical approaches to assess the degree of mtDNA mutations and repair capacity in oocytes.

## 10. Future Direction

The outcomes of ART in older women can be assessed through a range of clinical, biological, and demographic parameters [157]. Clinical indicators include the number and quality of oocytes retrieved, the fertilization rate, the number and quality of embryos reaching the blastocyst stage, the implantation rate, the confirmation of pregnancy via ultrasound evidence of a gestational sac, and the live birth rate per ART cycle. Biological and molecular parameters encompass hormonal levels, such as estradiol and progesterone during stimulation cycles, endometrial thickness, anti-Müllerian hormone levels, follicle-stimulating hormone levels, antral follicle count as visualized on ultrasound, and embryo genetic assessments, including preimplantation genetic testing for aneuploidy. Health and demographic factors, including the woman’s age, body mass index, comorbidities, and lifestyle behaviors such as smoking and alcohol consumption, further influence ART success. Moreover, follicular fluid, integral to oocyte development and maturation, serves as a biochemical mirror of the follicular microenvironment, wherein its constituent factors may serve as predictive indicators of ART outcomes. Recent review articles have explored the potential of evaluating oocyte and granulosa cell function by measuring markers of oxidative stress, cytokines, glycolysis, oxidative phosphorylation, metabolites, lipid profiles, autophagy, ferroptosis, and apoptosis in follicular fluid [19,158,159].

Unlike nuclear DNA, mtDNA lacks a mechanism for repairing double-stranded breaks. Instead, it has evolved mechanisms such as base excision repair for single-stranded breaks and mitophagy, which ensures the quality control of oocytes. Mitochondria were evolutionarily adapted to assume humans would undergo pregnancy and childbirth during only 20 to 30 years of a typical 50-year lifespan. Additionally, it was unforeseen that mechanisms to maintain mtDNA integrity would need to function effectively over such an extended lifespan. In this review, we emphasize the potential of real-time evaluation of mitochondrial DNA damage, repair, and replication mechanisms to improve the predictive accuracy of ART outcomes. Given the critical role of mitochondrial dysfunction, oxidative stress, mtDNA mutations, and compromised repair pathways in ovarian aging, identifying pharmacological interventions to target these molecular pathways is paramount in addressing ovarian aging. Direct assessment of mtDNA mutations in oocytes remains challenging, highlighting the pressing need for novel biomarkers capable of distinguishing between severe, irreversible damage and reversible damage based on the extent of mtDNA mutations. For instance, measuring ATP and NAD⁺ levels in follicular fluid and culture supernatants may offer valuable insights, as these biomarkers reflect mitochondrial activity crucial for oocyte function. Further studies are essential to determine whether the combined analysis of these biomarkers can significantly enhance ART outcomes.

## 11. Materials and Methods

### Search Strategy and Selection Criteria

A narrative review was conducted using the PubMed and Google Scholar databases to identify relevant studies published up to 30 September 2024, employing the keywords listed in Table 1. This review includes both human and animal data, with search terms combined using the Boolean operators AND and OR. Additionally, a manual search of reference lists from pertinent publications was performed. Eligible studies comprised original research articles in English and reference lists from review papers. Duplicate records, non-relevant literature, and non-English publications were excluded. Initial records were identified through electronic searches, and titles and abstracts were screened to exclude irrelevant studies after duplicates were removed. In the final eligibility assessment, full-text articles were reviewed to exclude studies lacking comprehensive data. The authors independently evaluated the relevance of the selected articles before proceeding with an exhaustive full-text review, resolving any uncertainties or disagreements through discussion. The study selection process is depicted in the flowchart in Figure 5, detailing the inclusion and exclusion criteria.

## Figures and Tables

**Figure 1 ijms-25-13144-f001:**
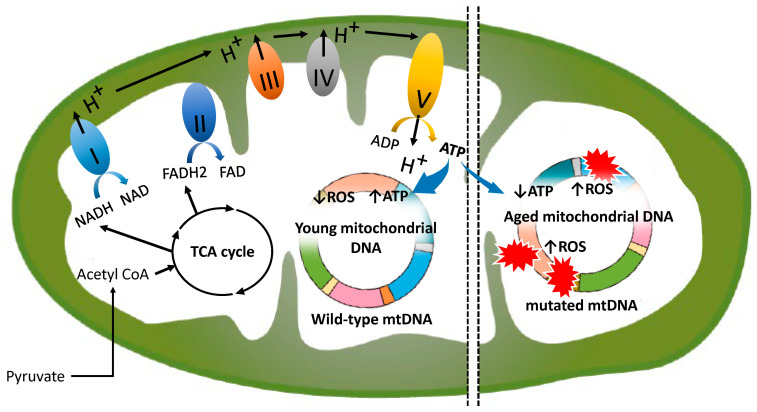
Mechanisms of the mitochondrial electron transport chain and its deterioration with aging. “I” to “V” denote the complexes of the mitochondrial electron transport chain. The colored ring indicates “mtDNA”. ROS-mediated mtDNA damage (red explosion marks) refers to “mutated mtDNA”. The left and right colored rings illustrate young and aged mitochondrial DNA, respectively.

**Figure 2 ijms-25-13144-f002:**
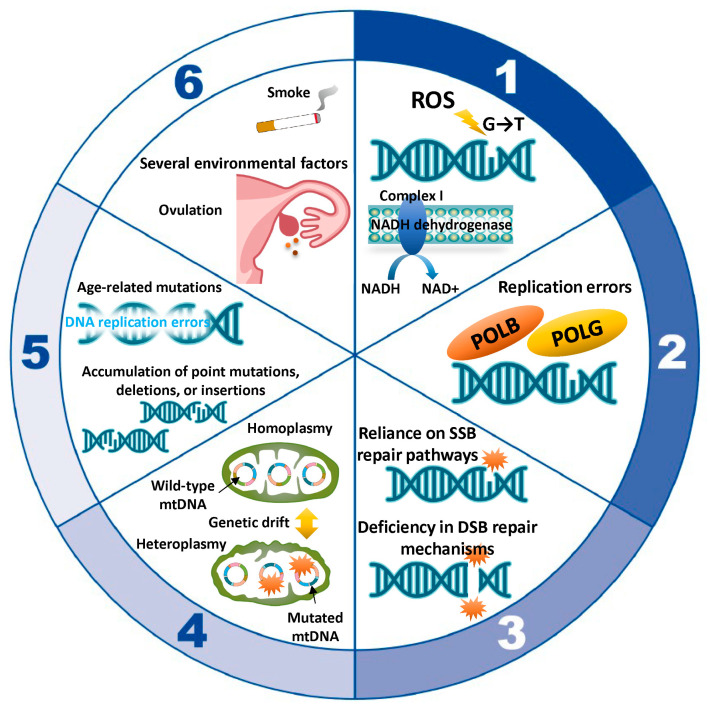
The molecular mechanisms of mtDNA mutations in oocyte aging. 1. ROS-induced damage: Oxidative stress can cause guanine (G) in DNA to undergo oxidation, resulting in its conversion to thymine (T). Complex I is particularly vulnerable to oxidative stress, and ROS-induced oxidation can impair its function, thereby disrupting cellular energy production. 2. Replication errors: The mitochondrial DNA polymerase, polymerase γ (POLG), is responsible for mtDNA replication, a process inherently more complex and prone to errors than the replication of nuclear DNA. Polymerase β (POLB) also plays a role in mitochondrial BER. 3. Mitochondrial DNA repair deficiencies: The repair mechanisms available for mtDNA are significantly more limited compared to those safeguarding nuclear DNA. The nucleotide excision repair, mismatch repair, and double-strand break repair pathways are less robust and not as extensively characterized as in the nucleus. 4. Replicative segregation and genetic drift: MtDNA is randomly allocated among daughter cells through replicative segregation. This randomness can result in the unpredictable dominance of either deleterious or neutral mutations. 5. Age-related accumulation of mutations: As individuals age, replication errors—such as point mutations, deletions, or insertions—accumulate in the mitochondrial genome. These mutations compromise mitochondrial function by disrupting the synthesis of proteins essential for the electron transport chain, reducing energy production, exacerbating oxidative stress, and contributing to the development of age-related diseases, including infertility. 6. Environmental factors: Ovulation-associated bleeding, smoking, and other environmental factors induce oxidative stress, which, in turn, promotes mutations in mtDNA. SSB, single-strand DNA break; DSB, double-strand DNA break.

**Figure 3 ijms-25-13144-f003:**
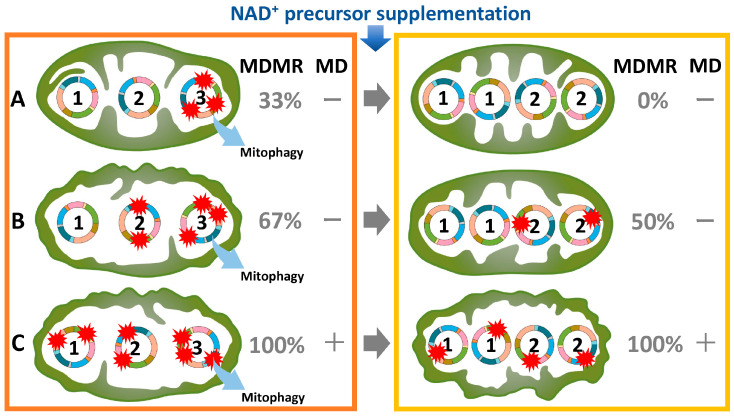
Mitochondrial dysfunction driven by the extent of mtDNA mutations (**left**) and the therapeutic potential of NAD⁺ precursor supplementation (**right**). Type A exhibits few mutations, type B has moderate mutations, and type C is characterized by numerous mutations. This diagram illustrates that each mitochondrion contains three mtDNA strands, labeled 1, 2, and 3 sequentially from left to right, with NAD⁺ precursor supplementation therapy resulting in a doubling of the mtDNA strand count. It should be noted that this depiction is a conceptual visualization intended to aid understanding of the mechanism and does not accurately reflect the actual biological system. Irreparable mtDNA mutations (mtDNA strand No. 3) are eliminated via mitophagy. Mitochondrial dysfunction is defined by the presence of mutations in 80% or more of the mtDNA. MDMR, mitochondrial DNA mutation rate; MD, mitochondrial dysfunction.

**Figure 4 ijms-25-13144-f004:**
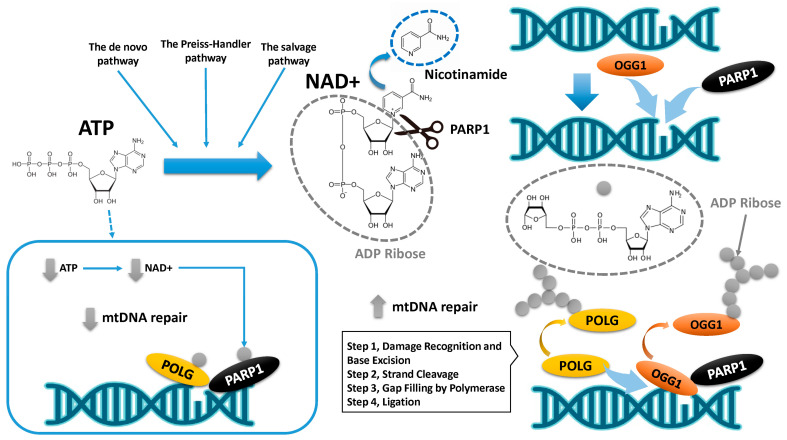
Mechanism of mtDNA repair by PARP1. The grey and blue dotted circles represent ADP-ribose and nicotinamide, respectively. The scissors represent the enzyme PARP1. The small grey circle indicates “ADP-ribose”. This figure illustrates only the involvement of NAD+ in the mechanism of mtDNA damage repair by PARP1. See the text for steps of BER involving PARP and polymerases.

**Figure 5 ijms-25-13144-f005:**
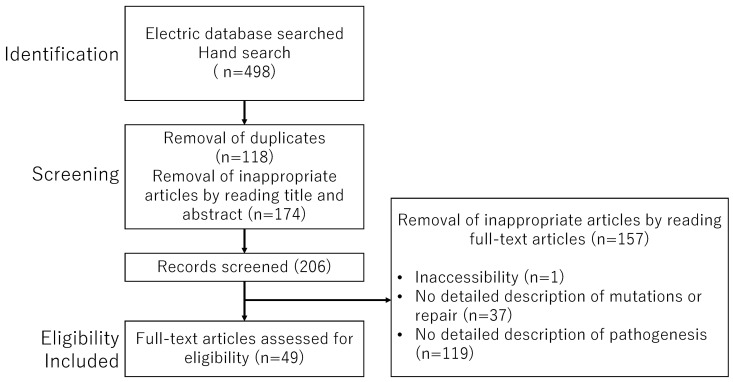
The flowchart outlines the study selection process.

**Table 1 ijms-25-13144-t001:** The keyword and search term combinations.

Search Mode	The Keyword and Search Term Combinations
Search term 1	mitochondria
Search term 2	aging
Search term 3	oocytes OR granulosa cells OR cumulus cells
Search term 4	mtDNA mutations OR mtDNA repair
Search term 5	oxidative stress OR reactive oxygen species OR ROS
Search term 6	nicotinamide adenine dinucleotide OR NAD OR NADH OR NADPH
Search term 7	poly ADP-ribose polymerase OR PARP1
Search	Search term 1 AND Search term 2 AND Search term 3
	Search term 1 AND Search term 2 AND Search term 3 AND Search term 4
	Search term 1 AND Search term 2 AND Search term 3 AND Search term 5
	Search term 1 AND Search term 2 AND Search term 3 AND Search term 6
	Search term 1 AND Search term 2 AND Search term 3 AND Search term 7

## Data Availability

No new data were created.

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
