# Peer review of "Mitochondrial DNA Damage and Its Repair Mechanisms in Aging Oocytes"

_ijms, 2024, doi:10.3390/ijms252313144_

Round 1

Reviewer 1 Report

Comments and Suggestions for Authors

Dear Authors

This review is relatively well-written. 

Here are minor comments

1. In the part of 2.4. Molecular mechanism underlying oocyte aging caused by mtDNA mutations, the contents of 1-6 were too short. Authors should add more sentences in detail. Furthermore, the numbering pattern of 1-6 seemed to be wrong. Please, check it. 

Author Response

Answer to the reviewers

ijms-3344770

Title: Mitochondrial DNA damage and its repair mechanisms in aging oocytes

Thank you and the reviewers for the thoughtful comments and helpful suggestions on our manuscript. We have carefully considered each of the comments, made every effort to address the concerns raised, and applied corresponding revisions to the manuscript. Additionally, we have carefully revised the manuscript to ensure that the text is optimally phrased and free from typographical and grammatical errors.

The detailed, point-by-point responses to the reviewer comments are given below, whereas the corresponding revisions are highlighted to our manuscript within the document (see blue letters).

We believe that our manuscript has been considerably improved as a result of these revisions, and hope that the revised manuscript is acceptable for publication in IJMS.

I would like to thank you once again for your consideration of our work and inviting me to submit the revised manuscript. I look forward to hearing from you.

With best regards,

Hiroshi Kobayashi, M.D., Ph.D.

E-mail: hirokoba@naramed-u.ac.jp

Reviewer 1

Comment 1:

This review is relatively well-written.

Here are minor comments

  1. In the part of 2.4. Molecular mechanism underlying oocyte aging caused by mtDNA mutations, the contents of 1-6 were too short. Authors should add more sentences in detail. Furthermore, the numbering pattern of 1-6 seemed to be wrong. Please, check it.

Response 1:

Thank you for your kind review.

See section 5, second paragraph: Subsection 5.1 is elaborated upon in greater depth in Sections 6 and 7. Subsection 5.2 is further reviewed in detail in Section 8. Given the limited understanding of mitochondrial DNA damage and its role in oocyte aging, we will provide only a concise discussion on this topic from 5.3 onwards. They have been renumbered.

Reviewer 2 Report

Comments and Suggestions for Authors

Kobayashi et al., in the present Review article decided to focus on the role of mitochondrial DNA (mtDNA) damage and relative repair mechanisms in aging oocytes. The authors sought to provide the reader with a brief summary on the current knowledge of age-related mitochondrial dysfunction and ovarian aging, the causes and mechanisms underlying mitochondrial DNA mutations and their potential repair, discussing potential therapeutic strategies to alleviate the impact of aging on oocytes.

The topic of the Review is interesting and scientifically relevant. Good references would be:

-PMID: 34091076

-PMID: 36068367

-PMID: 38355825

However, the Review must be improved since it is not well articulated, sometimes sounds repetitive and contains several inaccuracies.

For example:

In the “Results” section, “Normal mitochondrial function”, the authors imprecisely described oxidative phosphorylation (OXPHOS) and it is unclear if ROS are always genotoxic (please refer to PMID: 17848967). Figure 1 does not illustrate what the relative legend describes. Similarly: pag.4, lines 133-134.

Sometimes, the authors introduce a topic without enough background. As for example, at pag. 4, lines 149-151. Why do the macaques have those mutations? Due to aging? And compared to what?

Pag. 4, line 168: The authors introduce molecular mechanisms that are not mentioned and described.

Pag. 5, line185: This statement is confusing. Please refer to PMID: 19029901; PMID: 18223651; PMID: 20130269; PMID: 9500544.

Figure 2: This Figure does not provide the reader with the information required for “Molecular mechanisms of mtDNA mutations in oocytes aging”. Moreover, mitochondria do not possess a DNA repair mechanism for DNA double strand breaks. In the context of exogenous genotoxic agents, the authors considered only smoke.

Pages 6-7, paragraphs 1-6: The description provided is very limited and imprecise.

Pag. 7, paragraph 6: The authors should include and discuss the newly mitochondrial-localized DNA polymerase beta.

Pag. 7, lines 283 and 299: The statements in the two paragraphs contradict each other and are confusing.

Pag. 7, line 295: To which mechanisms the authors refer to?

Pag. 7, line 300: Please justify “certain mitochondrial mutations”. This statement is very vague.

Pag. 8, line 321: The authors should provide information/references about the “assumptions” in preparing the diagram.

Paragraph 2.4.3: This paragraph contains many inaccuracies and lacks important information about the main mechanism through which mitochondria safeguard their genome, as for example the Base Excision Repair mechanism.

Figure 4 is imprecise and must be improved.

Pag. 10: Sirtuins must be properly described in the context of mitochondria and mitochondrial DNA maintenance.

References are lacking, as for example:

- pag. 4, line 160

- pag. 8, lines 319, 320, 332

- pag. 9, line 350

- pag. 11, lines 424, 472

- pag. 12, lines 483, 504

Pag.3, line 99: How many mechanisms are known for OXPHOS?

Author Response

Answer to the reviewers

ijms-3344770

Title: Mitochondrial DNA damage and its repair mechanisms in aging oocytes

Thank you and the reviewers for the thoughtful comments and helpful suggestions on our manuscript. We have carefully considered each of the comments, made every effort to address the concerns raised, and applied corresponding revisions to the manuscript. Additionally, we have carefully revised the manuscript to ensure that the text is optimally phrased and free from typographical and grammatical errors.

The detailed, point-by-point responses to the reviewer comments are given below, whereas the corresponding revisions are highlighted to our manuscript within the document (see purple letters).

We believe that our manuscript has been considerably improved as a result of these revisions, and hope that the revised manuscript is acceptable for publication in IJMS.

I would like to thank you once again for your consideration of our work and inviting me to submit the revised manuscript. I look forward to hearing from you.

With best regards,

Hiroshi Kobayashi, M.D., Ph.D.

E-mail: hirokoba@naramed-u.ac.jp

Reviewer 2

Kobayashi et al., in the present Review article decided to focus on the role of mitochondrial DNA (mtDNA) damage and relative repair mechanisms in aging oocytes. The authors sought to provide the reader with a brief summary on the current knowledge of age-related mitochondrial dysfunction and ovarian aging, the causes and mechanisms underlying mitochondrial DNA mutations and their potential repair, discussing potential therapeutic strategies to alleviate the impact of aging on oocytes.

Comment 1:

The topic of the Review is interesting and scientifically relevant. Good references would be:

-PMID: 34091076

van der Reest J, Nardini Cecchino G, Haigis MC, Kordowitzki P. Mitochondria: Their relevance during oocyte ageing. Ageing Res Rev. 2021 Sep;70:101378. doi: 10.1016/j.arr.2021.101378.

-PMID: 36068367

Charalambous C, Webster A, Schuh M. Aneuploidy in mammalian oocytes and the impact of maternal ageing. Nat Rev Mol Cell Biol. 2023 Jan;24(1):27-44. doi: 10.1038/s41580-022-00517-3.

-PMID: 38355825

Leem J, Lee C, Choi DY, Oh JS. Distinct characteristics of the DNA damage response in mammalian oocytes. Exp Mol Med. 2024 Feb;56(2):319-328. doi: 10.1038/s12276-024-01178-2.

Response 1:

Thank you for providing the relevant references, we have used them as reference material. However, the paper by Leem et al. was not included as it is a review mainly on nuclear DNA repair.

Comment 2:

However, the Review must be improved since it is not well articulated, sometimes sounds repetitive and contains several inaccuracies.

Response 2:

In the manuscript, we have eliminated repetition as much as possible and corrected inaccuracies pointed out by reviewer 2.

Comment 3:

For example:

In the “Results” section, “Normal mitochondrial function”, the authors imprecisely described oxidative phosphorylation (OXPHOS) and it is unclear if ROS are always genotoxic (please refer to PMID: 17848967).

Response 3:

Reactive oxygen species (ROS) as by-products of mitochondrial respiration function as critical signaling molecules in preserving the health and homeostasis of organisms [33]; however, excessive ROS production generates cytotoxic and mutagenic free radicals, posing significant risks [6,26,28].

Comment 4:

Similarly: pag.4, lines 133-134.

Response 4:

The most critical consequence of age-related mitochondrial dysfunction is impaired ATP synthesis and mtDNA damage, which negatively influences chromosome segregation [46,47] and compromises embryonic development [48-50] (Figure 1).

"ROS-mediated" has been removed from this sentence.

Comment 5:

Figure 1 does not illustrate what the relative legend describes.

Response 5:

Regarding Figure 1, another reviewer also pointed this out, so we have redrawn it.

Comment 6:

Sometimes, the authors introduce a topic without enough background. As for example, at pag. 4, lines 149-151. Why do the macaques have those mutations? Due to aging? And compared to what?

Response 6:

The text has been corrected.

Arbeithuber et al. conducted an analysis of mitochondrial DNA (mtDNA) mutations in 30 Indian rhesus macaques, ranging in age from 1 to 23 years, stratified into four distinct age groups: <5 years, 5–10 years, 10–15 years, and 15–23 years [58]. Their findings revealed that de novo mtDNA mutations progressively accumulate with advancing age, predominantly manifesting as transitions rather than transversions [58].

Comment 7:

Pag. 4, line 168: The authors introduce molecular mechanisms that are not mentioned and described.

Response 7:

Another reviewer pointed this out, so we removed it and replaced it with the following text:

Evidence suggests that the majority of mtDNA mutations associated with aging are tran-sitions, whereas ROS-induced mutations are predominantly transversions [64]. The dominance of transitions over transversions in mtDNA mutations is noteworthy, as it highlights the biochemical processes underlying mutagenesis. Transitions are primarily linked to replication errors or repair mechanisms that favor nucleotide mispairing within the same chemical class [64]. Therefore, it is hypothesized that mtDNA replication errors become increasingly frequent with advancing age. On the other hand, ROS primarily in-duce transversion mutations, such as the conversion of guanine to 8-oxoguanine, which pairs with adenine and results in a G → T transversion [63]. This indicates that mitochon-dria possess an efficient capacity to repair mtDNA damage induced by oxidative stress [65].

Comment 8:

Pag. 5, line185: This statement is confusing. Please refer to PMID: 19029901; PMID: 18223651; PMID: 20130269; PMID: 9500544.

Response 8:

The text has been rewritten to make it easier to understand.

Mitochondria have developed mechanisms to minimize the intergenerational transmission of deleterious mtDNA mutations, a concept encapsulated by the bottleneck theory. The genetic bottleneck in mtDNA refers to the sharp reduction in mtDNA copy number during embryonic oogenesis, followed by amplification in subsequent stages [27,76,80,82,83,103-105]. While one consequence of this process is a reduction in the transmission of deleterious mtDNA mutations, the primary purpose and theoretical basis of the bottleneck extend beyond mutation reduction. The bottleneck amplifies stochastic differences in the proportions of mutant and wild-type mtDNA. This amplifies selection opportunities. If deleterious mutations dominate the reduced pool, these oocytes are less likely to develop or be viable. This is because irreparable mtDNA mutations are removed through mitophagy [106]. If wild-type mtDNA predominates, these oocytes have a better chance of propagating.

PMID: 19029901

Wai T, Teoli D, Shoubridge EA. The mitochondrial DNA genetic bottleneck results from replication of a subpopulation of genomes. Nat Genet. 2008 Dec;40(12):1484-8. doi: 10.1038/ng.258.

This paper is cited in Section 6.

PMID: 18223651

Cree LM, Samuels DC, de Sousa Lopes SC, Rajasimha HK, Wonnapinij P, Mann JR, Dahl HH, Chinnery PF. A reduction of mitochondrial DNA molecules during embryogenesis explains the rapid segregation of genotypes. Nat Genet. 2008 Feb;40(2):249-54. doi: 10.1038/ng.2007.63.

This paper has already been cited.

 PMID: 20130269

Wai T, Ao A, Zhang X, Cyr D, Dufort D, Shoubridge EA. The role of mitochondrial DNA copy number in mammalian fertility. Biol Reprod. 2010 Jul;83(1):52-62. doi: 10.1095/biolreprod.109.080887.

This paper is cited in Section 6.

PMID: 9500544

Larsson NG, Wang J, Wilhelmsson H, Oldfors A, Rustin P, Lewandoski M, Barsh GS, Clayton DA. Mitochondrial transcription factor A is necessary for mtDNA maintenance and embryogenesis in mice. Nat Genet. 1998 Mar;18(3):231-6. doi: 10.1038/ng0398-231.

This paper is cited in Section 8.

Comment 9:

Figure 2: This Figure does not provide the reader with the information required for “Molecular mechanisms of mtDNA mutations in oocytes aging”.

Moreover, mitochondria do not possess a DNA repair mechanism for DNA double strand breaks.

Response 9:

Subsections 5.1 to 5.6 correspond to numbers 1 through 6 in Figure 2, respectively, facilitating seamless cross-referencing for the reader. Figure 2 has been revised.

Comment 10:

In the context of exogenous genotoxic agents, the authors considered only smoke.

Response 10:

The following text was added:

Several environmental factors can induce mtDNA mutations by causing damage to the mitochondrial genome or disrupting its replication and repair processes. These factors include ultraviolet radiation, chemical toxins (pollutants, pesticides, tobacco smoke, and iron), radiation exposure, drugs and pharmaceuticals (chemotherapeutic agents, antibiotics, and endocrine-disrupting chemicals), and dietary and lifestyle factors (poor diet and alcohol consumption) [93]. Each of these factors can exacerbate the natural accumulation of mtDNA mutations over time, potentially contributing to aging and various mitochondrial and systemic diseases.

Comment 11:

Pages 6-7, paragraphs 1-6: The description provided is very limited and imprecise.

Response 11:

Subsections 5.1 through 5.6 served solely as concise overviews, while Subsections 5.1, 5.2, and 5.4 are elaborated upon in greater detail in Sections 7, 8, and 6, respectively.

Comment 12:

Pag. 7, paragraph 6: The authors should include and discuss the newly mitochondrial-localized DNA polymerase beta.

Response 12:

The following text was added:

Moreover, POLB plays a direct role in repairing oxidative lesions, such as 8-oxoguanine (8-oxoG), through its polymerase activity, ensuring mtDNA integrity [65]. POLB contrib-utes high-fidelity nucleotide incorporation during repair. This prevents mutagenic events that could arise from errors in repair synthesis. In addition to single-nucleotide repair, POLB can participate in long-patch BER, where a stretch of 2–10 nucleotides is replaced. Refer to Section 8 for a comprehensive overview of the mechanisms involved in mtDNA damage repair.

Comment 13:

Pag. 7, lines 283 and 299: The statements in the two paragraphs contradict each other and are confusing.

Response 13:

By line 299, do you mean line 289?

It has been amended as follows:

In humans, oocytes can remain arrested at prophase I for up to 50 years [6]. During this prolonged arrest, During the prolonged arrest lasting up to 50 years [6], mtDNA is susceptible to accumulating damage or replication errors, which may result in heteroplasmy or mtDNA deletions [58].

Comment 14:

Pag. 7, line 295: To which mechanisms the authors refer to?

Response 14:

We explained bottleneck theory in an easy-to-understand manner for readers.

Mitochondria have developed mechanisms to minimize the intergenerational transmission of deleterious mtDNA mutations, a concept encapsulated by the bottleneck theory. The genetic bottleneck in mtDNA refers to the sharp reduction in mtDNA copy number during embryonic oogenesis, followed by amplification in subsequent stages [27,76,80,82,83,103-105]. While one consequence of this process is a reduction in the transmission of deleterious mtDNA mutations, the primary purpose and theoretical basis of the bottleneck extend beyond mutation reduction. The bottleneck amplifies stochastic differences in the proportions of mutant and wild-type mtDNA. This amplifies selection opportunities. If deleterious mutations dominate the reduced pool, these oocytes are less likely to develop or be viable. This is because irreparable mtDNA mutations are removed through mitophagy [106]. If wild-type mtDNA predominates, these oocytes have a better chance of propagating.

Comment 15:

Pag. 7, line 300: Please justify “certain mitochondrial mutations”. This statement is very vague.

Response 15:

This implies heterozygous mtDNA mutations being transmitted to the blood and buccal cells of the offspring. It has been revised as follows because it was misleading.

The observed correlation between advanced maternal age at fertilization and an elevated number of heteroplasmies in the offspring implies that the transmission frequency of mutations increases with maternal age [90,109].

Comment 16:

Pag. 8, line 321: The authors should provide information/references about the “assumptions” in preparing the diagram.

Response 16:

This diagram illustrates that each mitochondrion contains three mtDNA strands, labeled 1, 2, and 3 sequentially from left to right, with NAD⁺ precursor supplementation therapy resulting in a doubling of the mtDNA strand count. It should be noted that this depiction is a conceptual visualization intended to aid understanding of the mechanism and do not accurately reflect the actual biological system.

Comment 17:

Paragraph 2.4.3: This paragraph contains many inaccuracies and lacks important information about the main mechanism through which mitochondria safeguard their genome, as for example the Base Excision Repair mechanism.

Response 17:

We have added information on the main mechanisms by which mitochondria protect their genomes, including the base excision repair mechanism.

We initially present a summary of the mechanisms governing mtDNA repair in oo-cytes (Figure 4). Among the mtDNA damage repair mechanisms, the BER pathway is critical for maintaining mtDNA integrity, addressing damage caused by ROS and other genotoxic insults [134,135]. The roles of Poly(ADP-ribose) polymerase (PARP) and polymerase in this process are as follows: PARP-1/OGG1 binary complexes detect single-strand breaks (SSBs) or nicks caused by damaged bases that have been excised. Once activated, PARP1 catalyzes the addition of poly(ADP-ribose) (PAR) chains to itself and nearby proteins using NAD⁺ as a substrate [136,137]. PARP-1 catalyzes the cleavage of NAD⁺ into nicotinamide and ADP-ribose. This modification recruits and stabilizes BER proteins at the site of damage, enhancing repair efficiency. Mitochondrial BER relies on specific DNA polymerases that perform vital roles in the repair mechanism. POLG is the primary mitochondrial DNA polymerase and plays a dual role in replication and repair. During BER, after the damaged base is excised by a glycosylase and the resulting abasic site is processed by an AP endonuclease, POLG fills the gap by incorporating the correct nucleotide. It also exhibits proofreading activity, ensuring high fidelity during gap filling. Emerging evidence suggests that additional polymerases, such as POLB, may transiently localize to mitochondria under stress [65]. Therefore, PARP activity and polymerase function are tightly regulated to maintain mitochondrial integrity. Dysfunction in these proteins, depletion of NAD⁺ levels, or introduction of replication errors can lead to impaired BER, mtDNA mutations, and mitochondrial dysfunction.

Comment 18:

Figure 4 is imprecise and must be improved.

Response 18:

Part of Figure 4 has been revised. However, this figure illustrates only the involvement of NAD+ in the mechanism of mtDNA damage repair by PARP1. See the text for steps of BER involving PARP and polymerases.

Comment 19:

Pag. 10: Sirtuins must be properly described in the context of mitochondria and mitochondrial DNA maintenance.

Response 19:

We added the following to the second paragraph of Section 8:

Second, NAD⁺ also supports mitochondrial sirtuins, enzymes essential for metabolic regulation, genomic stability, DNA repair, inflammation control, and longevity [140]. Sirtuins modulate mitochondrial enzymes involved in DNA repair processes such as BER. Sirtuins catalyze lysine deacetylation, coupling it to NAD⁺ hydrolysis, generating O-acetyl-ADP-ribose and nicotinamide [141]. For example, SIRT1 has been proposed to enhance the expression of nuclear respiratory factor 1 (NRF-1), NRF-2, and mitochondrial transcription factor A (TFAM), a key protein responsible for stabilizing mtDNA and promoting its replication and transcription [142], via its deacetylation of peroxisome proliferator-activated receptor gamma coactivator 1-alpha (PGC-1α). Moreover, SIRT3 activates mitochondrial antioxidant enzymes like superoxide dismutase 2 (SOD2) and glutathione peroxidase, reducing oxidative damage to mtDNA [143]. This helps maintain the integrity of the mitochondrial genome. Additionally, sirtuins rely on NAD⁺ for their enzymatic activity, linking their function to cellular energy status and mitochondrial metabolism. Therefore, elevated NAD⁺ levels enhance sirtuin activity, promoting efficient mtDNA repair, replication, and transcription. Furthermore, sirtuins play a role in mitophagy [144], thereby removing mitochondria with extensively mutated or damaged mtDNA.

Comment 20:

References are lacking, as for example:

Response 20:

I cannot comprehend the specific page and line numbers indicated by the reviewer. The references have already been given. We are concerned that the line counts in the reviewer's paper may differ from the line counts in our original paper.

- pag. 4, line 160

ROS primarily induce transversion mutations, such as the conversion of guanine to 8-oxoguanine, which pairs with adenine and results in a G → T transversion [63].

- pag. 8, lines 319, 320,

Page 8, lines 319 and 320 are the title of Figure 3.

- pag. 8, line 332

Mutations were detected in the mtDNA of 28% of oocytes and 66% of cumulus cells, demonstrating a markedly lower mutation burden in germ cells relative to somatic cells [111,112].

- pag. 9, line 350

The concomitant loss of numerous genes can disrupt ATP synthesis and compromise energy metabolism. Studies on cloned bovine oocytes suggest that mtDNA deletions increase with age [97], though human studies present conflicting evidence regarding age-related accumulation of such deletions [116,118-120].

- pag. 11, lines 424, 472

Line 424 is at the bottom of page 10. The text on line 472 has been deleted.

- pag. 12, lines 483, 504

As mtDNA damage accumulates and repair capacity becomes overwhelmed or impaired, mutations continue to increase with age.

Therefore, the administration of NMN requires careful consideration of dosage, duration, delivery method, and the extent of pre-existing mtDNA mutations.

Since these are summaries, I don't think references are necessary.

Reviewer 3 Report

Comments and Suggestions for Authors

In the current review, the authors summarized the knowledge of the effects of a woman’s age on mitochondrial DNA damage and repair and its correlation with ART efficiency. In general, it is an interesting and informative review. The quality of this review could be improved if the authors not just listed data but also provided their opinions/views on these findings, their importance, and future directions. Currently, it looks like just statements of facts lacking interpretations/discussions.

The review is very long and contains multiple repeats of the same/similar information. It will be easier for the readers if the authors shorten and reorganize the material.

The English of the manuscript could be improved.

Suggestions:

This is a review. It should not be divided into Introduction, Results, Materials, and Discussion. This is all published data.

Lines 29-30: This statement is not based on facts; it is only an assumption. In addition, in the previous centuries, humans had a much shorter lifespan. There is no discussion of the negative effects of numerous pregnancies on women’s health including deaths during childbirth. It is still very high in some populations despite the achievements of modern medicine. Please revise.

Figure 1. It is unclear if the young or old mitochondrion is depicted here. Maybe show the comparison between young and old?

Replace Discussion with Conclusion and Future Direction. What would you like to know in the future to improve ART efficiency? What could be the ideal treatment in Vivo, before oocyte collection, and in Vitro during ART? This should not be long (one to two paragraphs) but informative and meaningful.

Comments on the Quality of English Language

The English of the manuscript could be improved.

Author Response

Answer to the reviewers

ijms-3344770

Title: Mitochondrial DNA damage and its repair mechanisms in aging oocytes

Thank you and the reviewers for the thoughtful comments and helpful suggestions on our manuscript. We have carefully considered each of the comments, made every effort to address the concerns raised, and applied corresponding revisions to the manuscript. Additionally, we have carefully revised the manuscript to ensure that the text is optimally phrased and free from typographical and grammatical errors.

The detailed, point-by-point responses to the reviewer comments are given below, whereas the corresponding revisions are highlighted to our manuscript within the document (see blue letters).

We believe that our manuscript has been considerably improved as a result of these revisions, and hope that the revised manuscript is acceptable for publication in IJMS.

I would like to thank you once again for your consideration of our work and inviting me to submit the revised manuscript. I look forward to hearing from you.

With best regards,

Hiroshi Kobayashi, M.D., Ph.D.

E-mail: hirokoba@naramed-u.ac.jp

Reviewer 3

Comment 1:

In the current review, the authors summarized the knowledge of the effects of a woman’s age on mitochondrial DNA damage and repair and its correlation with ART efficiency. In general, it is an interesting and informative review. The quality of this review could be improved if the authors not just listed data but also provided their opinions/views on these findings, their importance, and future directions. Currently, it looks like just statements of facts lacking interpretations/discussions.

The review is very long and contains multiple repeats of the same/similar information. It will be easier for the readers if the authors shorten and reorganize the material.

The English of the manuscript could be improved.

Response 1:

We have tried to eliminate duplication as much as possible, so please review again.

We have carefully proofread the English text, but is it okay?

Comment 2:

Suggestions:

This is a review. It should not be divided into Introduction, Results, Materials, and Discussion. This is all published data.

Response 2:

We rewrote this review article according to your instructions.

Comment 3:

Lines 29-30: This statement is not based on facts; it is only an assumption. In addition, in the previous centuries, humans had a much shorter lifespan. There is no discussion of the negative effects of numerous pregnancies on women’s health including deaths during childbirth. It is still very high in some populations despite the achievements of modern medicine. Please revise.

Response 3:

It has been rewritten as follows:

Our female hunter-gatherer ancestors devoted a significant portion of their reproductive years to pregnancy and lactation [1]. However, they were also subject to the adverse health outcomes associated with excessive pregnancies, including maternal and fetal mortality during childbirth.

Comment 4:

Figure 1. It is unclear if the young or old mitochondrion is depicted here. Maybe show the comparison between young and old?

Response 4:

Young mitochondria, depicted in the center of Figure 1, exhibit enhanced ATP synthesis and attenuated ROS production. Conversely, aged mitochondria, shown on the right, dis-play impaired ATP synthesis, mitochondrial dysfunction, elevated ROS production, and consequent mitochondrial DNA damage. Thus, mitochondria in younger cells predominantly harbor wild-type mtDNA, whereas those in aged cells exhibit a higher proportion of mutated mtDNA.

Figure 1 has been revised.

The left and right colored rings illustrate young and aged mitochondrial DNA, respectively.

Comment 5:

Replace Discussion with Conclusion and Future Direction. What would you like to know in the future to improve ART efficiency? What could be the ideal treatment in Vivo, before oocyte collection, and in Vitro during ART? This should not be long (one to two paragraphs) but informative and meaningful.

Response 5:

As instructed, we divided the “discussion” into “conclusions” and “future directions.”

The initial paragraph of the “Future Direction” section evaluates the current achievements and potential advancements of ART for older women, while the subsequent paragraph encapsulates future perspectives derived from the findings presented in this review.

Comment 6:

Comments on the Quality of English Language

The English of the manuscript could be improved.

Response 6:

We have carefully proofread the English text, but is it okay?

Round 2

Reviewer 2 Report

Comments and Suggestions for Authors

I appreciate the efforts of the authors to address the major concerns previously raised. I still find inaccuracies mainly regarding mitochondrial DNA repair mechanisms. I encourage the authors to read with more criticism the literature about this topic. Overall, the current version has improved sufficiently for publication.

Author Response

Comments and Suggestions for Authors
I appreciate the efforts of the authors to address the major concerns previously raised. I still find inaccuracies mainly regarding mitochondrial DNA repair mechanisms. I encourage the authors to read with more criticism the literature about this topic. Overall, the current version has improved sufficiently for publication.

Response
I reviewed a paper on the mechanisms of damage repair in nuclear and mitochondrial DNA, summarizing its findings in this article. While the repair mechanisms of nuclear DNA are comprehensively outlined in several articles, those pertaining to mitochondrial DNA appear to remain underdeveloped. You mentioned, "I still find inaccuracies mainly regarding mitochondrial DNA repair mechanisms." If you could specify the inaccuracies, I would be keen to address them. Alternatively, would it be acceptable to leave the text as is without further revisions?

Reviewer 3 Report

Comments and Suggestions for Authors

The authors addressed most of my critiques. The review was significantly improved.

Suggestion:

Lines 345-346: “Mitochondria have developed mechanisms to minimize the intergenerational transmission of deleterious mtDNA mutations”. Cells, oocytes, or evolution, not mitochondria developed this mechanism.

Author Response

Comments and Suggestions for Authors
Suggestion:
Lines 345-346: “Mitochondria have developed mechanisms to minimize the intergenerational transmission of deleterious mtDNA mutations”. Cells, oocytes, or evolution, not mitochondria developed this mechanism. 

Response
Thank you for pointing that out. Since we are explaining about oocytes here, I have corrected it to "Oocytes."